# LEARNING INVARIANCE WITH COMPACT TRANSFORMS

**Anna T. Thomas**\*, **Albert Gu**\*, **Tri Dao**\*, **Atri Rudra**† **& Christopher Ré**\*
\* Stanford University, {thomasat,albertgu,trid,chrismre}@cs.stanford.edu
† University at Buffalo, SUNY, atri@buffalo.edu

## ABSTRACT

The problem of building machine learning models that admit efficient representations and also capture an appropriate inductive bias for the domain has recently attracted significant interest. Existing work for compressing deep learning pipelines has explored classes of structured matrices that exhibit forms of shift-invariance akin to convolutions. We leverage the displacement rank framework to automatically learn the structured class, allowing for adaptation to the invariances required for a given dataset while preserving asymptotically efficient multiplication and storage. In a setting with a small fixed parameter budget, our broad classes of structured matrices improve final accuracy by 5–7% on standard image classification datasets compared to conventional parameter constraining methods.

## 1 INTRODUCTION

State of the art deep neural networks often require billions of parameters, necessitating reduced parameterization in resource-constrained settings. Among the broad families of approaches for learning more compact models, one line involves constraining the layers with some form of dense structure and learning directly over the parameterization of this structure. Much previous work in this area has explored classes of matrices that implicitly encode a shift-invariant structure (Cheng et al., 2015; Ding et al., 2017; Sindhwani et al., 2015). While their explicit aim is to accelerate training and reduce memory costs, this structure also introduces an inductive bias that is well-suited for image and audio data. Convolutional networks (LeCun et al., 1998) have demonstrated that leveraging such inductive bias, which forms a rich prior (Ulyanov et al., 2017), is essential to learning representations that are both fast and high quality. We aim to learn such priors or invariances from data rather than handcrafting them, while retaining the efficiency of previous approaches.

Our approach uses *low displacement-rank* (LDR) structured matrices, which decouple invariance and compactness. In fact, this formulation was originally used to define matrices with *shift invariance* structure (Kailath et al., 1979). LDR matrices encode invariance through two sparse displacement operators and control compactness through a low-rank remainder matrix. Unlike schemes for sparse or repeated parameters, the structured matrix approach produces a dense and regular structure in the final weight matrix with guaranteed compression (usually from quadratic to linear space). Because of their strong regularity and constraints, these matrices provide an opportunity to impose structure based on prior knowledge of how they might interact with the specific task or data. Previous work on LDR matrices (Sindhwani et al., 2015) used fixed and specialized displacement operators, as fast multiplication algorithms were only known for limited classes of operators. By employing recent progress in structured dense matrix multiplication (De Sa et al., 2018) that allows fast multiplication and compact storage of a much larger class of structured matrices, we learn over both the low-rank component and the displacement operators. We use constructions that unify and extend many of the previously considered structured classes and achieve improved performance on downstream tasks, while also seeking to explain what makes this family of constructions effective.

**Background on Displacement Rank.** The displacement approach is a broad method for representing structure that was originally used to describe matrices exhibiting shift invariance that are almost Toeplitz. It represents a structured matrix $\mathbf{A}$ through a *displacement operator* $\nabla_{\mathbf{M},\mathbf{N}}$ defining a linear map $\mathbf{A} \mapsto \mathbf{MA} - \mathbf{AN}$ on matrices, and a *low-rank residual* $\mathbf{R}$, so that if $\mathbf{MA} - \mathbf{AN} = \mathbf{R}$ then $\mathbf{A}$ can be manipulated solely through the compressed representation $(\mathbf{M}, \mathbf{N}, \mathbf{R})$. Many important classes of structured matrices such as Vandermonde and Cauchy matrices have been shown to satisfy a displacement property (Appendix A). However, conventionally a given family of matrices is described with respect to a fixed displacement operator $\mathbf{M}, \mathbf{N}$, and parameterized only by the remainder $\mathbf{R}$. In particular, the application of displacement rank in machine learning has been confined to the Toeplitz-like matrices, which fix $\mathbf{M}, \mathbf{N}$ to be *shift* or *cycle* operators (see Definition 1).

## 2 CAPTURING INDUCTIVE BIAS BY LEARNING DISPLACEMENT OPERATORS

Although seldom mentioned, the use of displacement rank to represent functions has an interpretable invariance-preserving or perturbation-resisting effect. For a feature map represented as a Toeplitz-like matrix, its displacement property involving shift operators implies that **shifting the input to the feature map yields a shifted output**. More generally, the compressed displacement representation $(\mathbf{M}, \mathbf{N}, \mathbf{R})$ can be viewed as decomposing into two parts: the displacement operator $\nabla_{\mathbf{M}, \mathbf{N}}$ models high-level structures and invariances in the data, and the remainder $\mathbf{R}$ performs more fine-tuned fitting like a standard low-rank approximation. Therefore learning a broader class of operators allows for modeling more general structures. Leveraging this insight as well as recent algorithms (De Sa et al., 2018), we construct simple neural nets with LDR layers where the displacement operators include variable parameters to be learned automatically from data. The resulting weight matrices still retain near-linear time and space properties (Appendix A).

**Scaled-cycle operators.** We first investigate a straightforward generalization of Toeplitz-like matrices, which were previously examined in deep learning, and for a small parameter budget outperformed other compression approaches (Sindhwani et al., 2015).

**Definition 1.** For $f \in \mathbb{R}$, let $\mathbf{Z}_f$ denote the $f$-unit circulant matrix $\begin{bmatrix} 0_{1 \times (n-1)} & f \\ I_{n-1} & 0_{(n-1) \times 1} \end{bmatrix}$. The case $\mathbf{Z}_{\pm 1}$ is used to define the Toeplitz-like matrices (see Appendix A, Table 1), and corresponds to the *cycle matrix* (Appendix A.2, Figure 3a). These matrices generally represent shifts or cycles. For a vector $\mathbf{x} \in \mathbb{R}^n$, define $\mathbf{Z}_{\mathbf{x}} = \operatorname{diag}(\mathbf{x})\mathbf{Z}_1$ to be the matrix with the same sparsity pattern as $\mathbf{Z}_1$ but parameterized by $\mathbf{x}$ (Appendix A.2, Figure 3b).

This class with "scaled-cycle" operators aligns with our goal of learning inductive bias (i.e. parameters of the displacement operator), as they are the most natural extension to Toeplitz-like matrices where we can observe the effects of learning over progressively more parameters in the operator. Surprisingly, this class also includes other previously considered models that were not realized to have low displacement rank (Appendix A.3). This emphasizes the representative power of our new structures, while our results here also provide insights into such previous work.

**Tridiagonal-plus-corner operators.** We next consider a broader class of matrices, with operators consisting of tridiagonal matrices (Appendix A.2, Figure 3c). This class is extremely rich, encompassing the classic classes of structured matrices (Toeplitz, Vandermonde, etc.), all previously studied forms of displacement rank (De Sa et al., 2018), the practical **ACDC** layer of Moczulski et al. (2016) (see Appendix A.3), and even normal low rank matrices (by setting $\mathbf{M} = \mathbf{I}$ and $\mathbf{N} = \mathbf{0}$).

## 3 EMPIRICAL EVALUATION

We follow the experimental setting of Chen et al. (2015) and Sindhwani et al. (2015) by testing a very compact architecture, a single hidden layer neural network, on a challenging MNIST variant (Larochelle et al., 2007) and a grayscale version of CIFAR-10. (Details in Appendix B.) In Sindhwani et al. (2015) it was shown that for comparable numbers of parameters, Toeplitz-like structured transforms significantly outperform other approaches such as Random Edge Removal, Dark Knowledge, HashedNets, and Low-rank Decomposition (Hinton et al., 2015; Ciresan et al., 2011; Chen et al., 2015). Using this as a baseline, we focus on comparing different types of displacement structures and on how parameters of different types (e.g. in the displacement operator vs. in the low-rank remainder) contribute to performance.

**Results.** Figure 1 shows that for a constant total number of parameters, our classes with learned operators outperform the fixed classes — as well as an unconstrained model of the same dimensions (623290 parameters) on MNIST-noise, demonstrating the value of an appropriate inductive bias. In particular, tridiagonal-plus-corner outperforms the best fixed class by 7.2% on MNIST-noise and 4.9% on the grayscale version of CIFAR-10. We include other classic displacement rank types, which Sindhwani et al. (2015) and Zhao et al. (2017) indicate as an unexplored possibility.

Of particular note is the poor performance of low-rank matrices. As mentioned in Section 1, a fixed displacement operator $\nabla_{\mathbf{M}, \mathbf{N}}$ is a linear map on the space of matrices, meaning that a fixed class of low displacement rank such as the Toeplitz-like matrices has essentially the same parameterization, i.e. they are merely the low-rank matrices under a different basis. We hypothesize that the

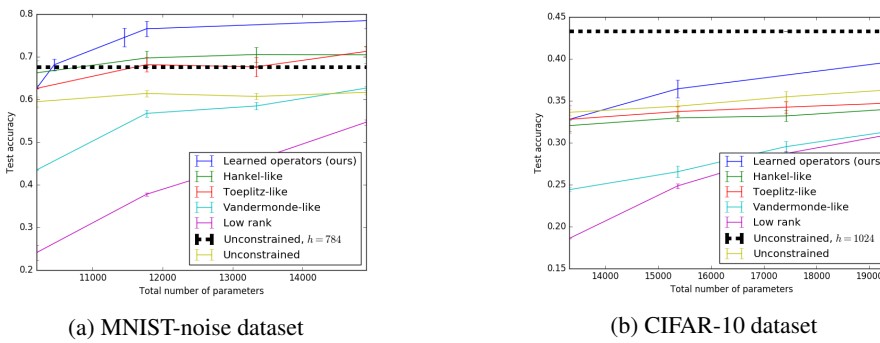

(a) MNIST-noise dataset

(b) CIFAR-10 dataset

Figure 1: Relationship between parameters vs. performance on a compact network.

main contribution to their marked performance difference is the effect of the displacement operator automatically taking care of certain patterns and invariances in the data. The improvement in the displacement rank classes, from Vandermonde-like to Toeplitz/Hankel-like to our new classes with learned operators, comes from more accurate representations of these invariances.

Note that the "Toeplitz-like" curve starts from the Toeplitz-like (rank 1) class and progressively increases the displacement **rank**. On the other hand, the "learned operators" curve also starts from the Toeplitz-like (rank 1) class but progressively relaxes the displacement **operator**. The results imply that on a per parameter basis, *the parameters governing high-level structure are much more effective at increasing performance than the standard rank parameters*.

**Interpretation of Learned Displacement Operators.** Figure 2a and 2b show the heat map of the weight matrix $\mathbf{W}$ ($784 \times 784$) trained on MNIST-noise dataset. The learned scaled-cycle operator is able to pick up regularity in the $28 \times 28$ input (flattened as a 784-vector), as the weight matrix exhibits much stronger grid-like periodicity of size 28 compared to the learned Toeplitz-like matrices. This periodicity might be attributed to learned patterns of the subdiagonal of the operator (i.e. the $\mathbf{x}$ vector in Definition 1), whose Discrete Fourier Transform as shown in Figure 2c peaks at 28 and 56. The operators of Toeplitz-like matrices, with a constant subdiagonal, are unable to pick up such patterns.

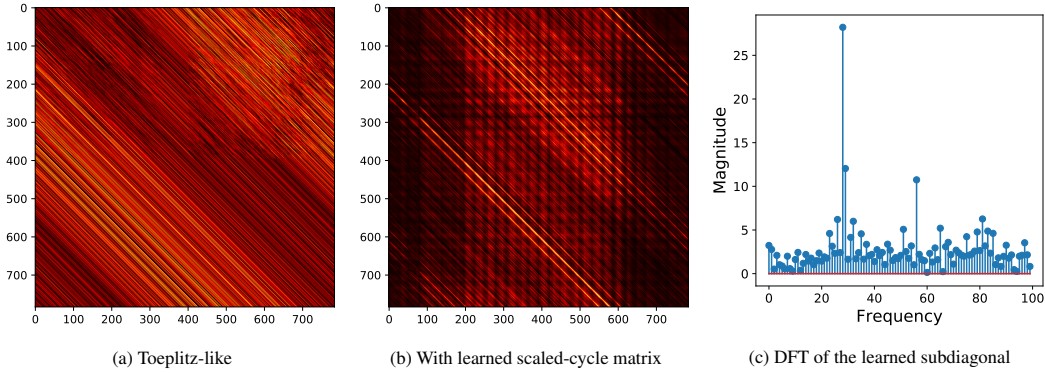

(a) Toeplitz-like

(b) With learned scaled-cycle matrix

(c) DFT of the learned subdiagonal

Figure 2: Visualization of learned weight matrix (a,b), and frequencies of the scaled-cycle subdiagonal (c).

## 4 CONCLUSION

Convolution layers in neural networks are extraordinarily effective at capturing the right inductive bias, as their filters use hand-crafted locality constraints that match the input modality (e.g. 2D image data). For the related problem of compressing dense fully-connected layers, many lines of work end up using similar but more domain-agnostic structured constructions, such as circulant and Toeplitz variants, that encode general dense convolutions. We further relax such Toeplitz (and related) structure and manage to recover domain-specific properties such as periodicity — akin to the type of structure imposed by specialized sparse convolution filters — while retaining fast multiplication and efficient storage properties.

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

| Structured Matrix $\mathbf{M}$ | $\mathbf{A}$ | $\mathbf{B}$ | Displacement Rank $r$ |
|---|---|---|---|
| Toeplitz | $\mathbf{Z}_1$ | $\mathbf{Z}_{-1}$ | $\leq 2$ |
| Hankel | $\mathbf{Z}_1$ | $\mathbf{Z}_0^T$ | $\leq 2$ |
| Vandermonde | $\mathrm{diag}(v)$ | $\mathbf{Z}_0$ | $\leq 1$ |
| Cauchy | $\mathrm{diag}(s)$ | $\mathrm{diag}(t)$ | $\leq 1$ |

Table 1: Commonly used classes of structured matrices.

# A    PROPERTIES OF DISPLACEMENT RANK

The displacement rank approach has been used to capture many important types of matrices frequently used in fields such as engineering and signal processing. Table 1 summarizes the displacement representations of these classic matrix families.

## A.1    CONNECTION BETWEEN DISPLACEMENT RANK AND INVARIANCE

Consider a Toeplitz-like matrix $\mathbf{A}$ representing a feature map, so that it satisfies the equation $\mathbf{ZA} - \mathbf{AZ} = \mathbf{R}$ for some low rank $\mathbf{R}$ (where $\mathbf{Z}$ is a shift matrix, see Definition 1). For any input vector $\mathbf{x}$, the displacement equation implies that $\mathbf{A}(\mathbf{Zx}) = \mathbf{Z}(\mathbf{Ax}) + \mathbf{Rx}$. Up to some simple error term $\mathbf{Rx}$, this means that shifting the input to the feature map yields a shifted output.

**Proposition 1.** *If $\mathbf{MA} - \mathbf{AN} = \mathbf{R}$, then the displacement equation implies that $\mathbf{A}(\mathbf{N}x) = \mathbf{M}(\mathbf{A}x) - \mathbf{R}x$. Thus, up to some simple error term $\mathbf{R}x$, this means that we can recover the original input after some perturbation $\mathbf{N}$ of the input.*

## A.2    RECONSTRUCTION AND MULTIPLICATION FOR OUR CLASSES OF STRUCTURED MATRICES

In this work, we introduce broad classes of displacement structure which we then automatically learn over to better capture invariances and biases in the domain. The displacement operators for the Toeplitz-like class as well as our new classes are drawn in Figure 3.

$$\begin{bmatrix} 0 & 0 & \cdots & 1 \\ 1 & 0 & \cdots & 0 \\ \vdots & \vdots & \ddots & \vdots \\ 0 & \cdots & 1 & 0 \end{bmatrix} \qquad \begin{bmatrix} 0 & 0 & \cdots & x_0 \\ x_1 & 0 & \cdots & 0 \\ \vdots & \vdots & \ddots & \vdots \\ 0 & \cdots & x_n & 0 \end{bmatrix} \qquad \begin{bmatrix} b_0 & a_0 & \cdots & s \\ c_0 & b_1 & \ddots & \vdots \\ \vdots & \ddots & \ddots & a_{n-1} \\ t & \cdots & c_{n-1} & b_n \end{bmatrix}$$

(a) Cycle matrix                        (b) Scaled-cycle matrix                        (c) Tridiagonal (plus corners)

Figure 3: Operator matrices for Toeplitz-like (a) and our learned classes (b,c).

Multiplication algorithms for these classes have been described in detail in previous works.

First, we mention an old reconstruction formula that can be used to recover matrices satisfying a special property from their displacement representation. This method was also used in Sindhwani et al. (2015) for a very specific case.

**Theorem 1** (Pan & Wang (2003))**.** *If an $n \times n$ matrix $\mathbf{M}$ is such that $\nabla_{\mathbf{A},\mathbf{B}}[M] = \mathbf{GH}^T$ where $\mathbf{G} = [g_1, \cdots, g_r], \mathbf{H} = [h_1, \cdots, h_r]$ and the operators satisfy $\mathbf{A}^n = a I$ and $\mathbf{B}^n = b I$ for some scalars $a, b$, then $\mathbf{M}$ can be expressed as:*

$$M = \frac{1}{1-ab} \sum_{j=1}^{r} \mathrm{Krylov}(\mathbf{A}, g_j) \, \mathrm{Krylov}(\mathbf{B}^T, h_j)^T \tag{1}$$

For the matrices with low displacement rank with respect to scaled-cycle operators, we can utilize this reconstruction.

**Proposition 2.** $\mathbf{Z}_{\mathbf{x}}^n = aI$, where $a = \prod_{i=0}^{n-1} x_i$.

*Proof.* Define $\mathbf{D}_{\mathbf{x}} = \mathrm{diag}(\mathbf{x})$. Then note that $\mathbf{Z}_{\mathbf{x}} = \mathbf{Z}_1 \mathbf{D}_{\mathbf{x}}$, $\mathbf{Z}_{\mathbf{x}}^2 = [x_1 x_2 e_3 \quad x_2 x_3 e_4 \quad ... \quad x_{n-2} x_{n-1} e_n \quad x_0 x_1 e_1 \quad x_0 x_1 e_2] = \mathbf{Z}_{\mathbf{x}} \mathbf{Z}_1 \mathrm{diag}(\mathbf{x})$, and so on. At each step, we shift the columns of $\mathbf{Z}_{\mathbf{x}}$ left and scale each column by the corresponding entry of $\mathbf{x}$. Thus, after multiplying $\mathbf{Z}_{\mathbf{x}}$ by itself $n-1$ times, all columns have been shifted left $n-1$ times (producing the sparsity pattern of $\mathbf{I}_n$) and have been scaled by $\prod_{i=0}^{n-1} x_i$. $\square$

The more general case of tridiagonal-plus-corner displacement operators can be handled by the general algorithm of De Sa et al. (2018). In this work, for simplicity and ease of implementation we approximate them using the same reconstruction formula (1).

### A.3 Subsuming the ACDC Construction

Moczulski et al. (2016) used the abstraction of "Structured Efficient Linear Layers" to describe a line of work on using structured matrices to compress fully-connected neural network layers, encompassing circulant weight matrices (Cheng et al., 2015), low-distortion projections (Yang et al., 2015), and so on. The novel construction that Moczulski et al. (2016) introduced were structured layers with the form $\mathbf{AFDF}$, where $\mathbf{A}$, $\mathbf{D}$ are diagonal matrices and $\mathbf{F}$ is the Discrete Fourier Transform. This turns out to have a displacement property, which is a consequence of the multiplicative closure property of displacement rank:

**Proposition 3** (Pan (2012)). *If* $\mathbf{MA} - \mathbf{AN} = 0$ *and* $\mathbf{NB} - \mathbf{BP} = 0$, *then* $\mathbf{M}(\mathbf{AB}) - (\mathbf{AB})\mathbf{P} = 0$.

Using the fact that the Fourier Transform $\mathbf{F}$ diagonalizes the cycle matrix $\mathbf{Z}_1$, and applying this property, shows that these matrices have displacement rank 0 with respect to a displacement operator consisting of one scaled-cycle matrix $\mathbf{Z}_{\mathbf{x}}$ and one cycle matrix $\mathbf{Z}_1$.

Furthermore, Moczulski et al. (2016) considers a more practical version $\mathbf{ACDC}$, which is the approximation of the first construction where $\mathbf{F}$ is replaced with the Discrete Cosine Transform $\mathbf{C}$. Using the fact that the DCT diagonalizes the Chebyshev polynomials's Jacobi matrix implies that this construction also satisfies a displacement property, but with respect to tridiagonal matrices.

## B Experimental Details

For the experiments we use a neural network with a single hidden layer. We set the learning rate to $10^{-3}$ and momentum parameter to 0.9 for all methods. In Figures 1, we compare the performance of our learned classes with the fixed classes Toeplitz-like, Hankel-like, Vandermonde-like, and low rank. We also compare with an unconstrained weight matrix of the same dimensions of the structured classes (denoted "unconstrained, $h = n$", where $n$ is the size of the input in Figure 1), as well as an unconstrained weight matrix where the number of hidden units is adjusted to yield the same total number of parameters (denoted "unconstrained" in Figure 1). The MNIST-noise dataset we test on was constructed via adding correlated pixel noise sampled from a zero-mean multivariate Gaussian distribution. Larochelle et al. (2007) construct six versions of this dataset with varying levels of background pixel correlation, and we test on the most challenging variant.

| Method | Number of Parameters | Mean Test Accuracy (Standard Deviation) |
|---|---|---|
| Learned tridiagonal+corner, $r = 1$ | 14904 | 0.784 (0.018) |
| Learned scaled-cycle, $r = 1$ | 11770 | 0.765 (0.018) |
| Toeplitz-like, $r = 4$ | 14906 | 0.712 (0.012) |
| Hankel-like, $r = 4$ | 14906 | 0.704 (0.019) |
| Unconstrained | 623290 | 0.6761 (0.002) |
| Vandermonde-like, $r = 4$ | 15690 | 0.626 (0.005) |
| Low rank, $r = 4$ | 14906 | 0.546 (0.006) |

Table 2: Performance on MNIST-noise dataset.

| Method | Number of Parameters | Mean Test Accuracy (Standard Deviation) |
|---|---|---|
| Unconstrained | 1059850 | 0.4326 (0.003) |
| Learned tridiagonal+corner, $r = 1$ | 19464 | 0.3966 (0.001) |
| Learned scaled-cycle, $r = 1$ | 15370 | 0.36406 (0.010) |
| Toeplitz-like, $r = 4$ | 19466 | 0.3472 (0.011) |
| Hankel-like, $r = 4$ | 19466 | 0.34005 (0.002) |
| Vandermonde-like, $r = 4$ | 20490 | 0.31405 (0.002) |
| Low rank, $r = 4$ | 19466 | 0.311 (0.002) |

Table 3: Performance on CIFAR-10 dataset.

**Rectangle Dataset.** We also provide an example of a case where the learned operators do not exceed the performance of the fixed classes. In this dataset, created by Larochelle et al. (2007), the task to classify, based on a binary image of a rectangle, whether the length is larger than the width. On this dataset, unlike the others, expanding the class beyond Toeplitz-like only reduces performance. We hypothesize this is because the necessary invariances for success on this simple dataset are already contained within the Toeplitz-like class, and further expansion of the class leads to overfitting.

| Method | Number of Parameters | Mean Test Accuracy (Standard Deviation) |
|---|---|---|
| Toeplitz-like, $r = 4$ | 14906 | 0.992 (0.003) |
| Learned scaled f-unit-circulant, $r = 1$ | 11770 | 0.989 (0.003) |
| Learned tridiagonal+corner, $r = 1$ | 14904 | 0.988 (0.009) |
| Unconstrained | 623290 | 0.953 (0.001) |

Table 4: Performance on rectangles dataset.

