# OpenReview forum: "Learning Invariance with Compact Transforms"
_ICLR.cc/2018/Workshop — Accept_

### Official Review · AnonReviewer1 · 2018-03-09
**generalized low rank structure for compression and guiding inductive bias in neural nets**

**Rating:** 8
**Confidence:** 3

**Review:**

The authors significantly generalize previous work in compressing weight matrices in neural nets by introducing the use of low displacement rank matrices. Prior works considered low-rank matrices and certain classes of structured matrices, including toeplitz, circulant, vandermonde, which are instances of LDR matrices.

The novel contribution of this work is to realize that this more general class of matrices can be used to decrease the model size, and to demonstrate that the inductive biases gained thereby can be useful. I recommend it be accepted, as I think this is an interesting and useful formalization for practitioners, both from the point of view of (potential) efficiency, and from its ability to learn and expose inductive biases --- e.g. in the best case one could imagine that learning the correct LDR structures might lead to improved performance on certain classes of non-image data in the same way that the form of the matrices in CNNs (in combination with structure of the connections) has proven to be quite useful for image data.

Pros:
- The authors unify and generalize a large number of weight matrix structures previously used to compress and guide inductive bias in neural networks, and demonstrate that learning the correct LDR structure is feasible and can lead to improved performance over previous such fixed structures.

Cons:
- The authors do not provide timing information to show that learning with LDR matrices can be accomplished efficiently as they claim
- The description of Figure 1 in Section 3 and the figure itself does not seem to match: shouldn't h in the figure be 623290?- - What does the Krylov(...) notation in Theorem 1 mean: how can one use this to compute a matrix-vector multiply?
- I would like to have seen a discussion of the implications of the Zhao et al. 2017 ICML paper on theoretical guarantees for using LDR matrices as weights to the approach suggested by the authors.

---

### Official Review · AnonReviewer3 · 2018-03-10
**Seems to have good results**

**Rating:** 8
**Confidence:** 3

**Review:**

The paper discusses a new framework for encoding structure in neural net layers (extending basic ideas like sparsity, convolutional structure, and others discussed in Sindhwani et al.). Their "LDR" structure captures enough information, while still reducing the number of parameters to show a benefit. The actual details of how to do this, and the training, is not explained, but the results on some standard test problems show promise.

This seems a very topical paper, with interesting results, and fits well in the workshop track. It could lead to a quite interesting full paper.

---

### Official Review · AnonReviewer2 · 2018-03-10
**An interesting paper about constructing compressed neural nets with low displacement rank layers where the displacement operators are learned from the data.**

**Rating:** 8
**Confidence:** 3

**Review:**

The paper belongs to the body of work investigating structured matrices for constructing compressed neural nets. The main idea here is to rely on low displacement rank structured matrices that can be represented by a displacement operator and a low-rank residual. The authors show that by learning the displacement operator which models the invariances in the data, they outperform state of the art approaches based on structured matrices by improving the final accuracy by 5–7% on standard image classification datasets.

The idea is very seductive, and it seems to rely on a well-founded theoretical development. That said, I must confess that I've been a little bit frustrated to not know more on how learning the displacement operator is concretely performed. The paper mentions that this is done by considering it as a parameter, but how the gradient is computed is not clear. It is probably not a hard task, but I recommend the authors to provide more details in the appendix.

---

### Public Comment · ~Oriol_Vinyals1 · 2018-02-17
**Please Fix Length**

Your paper violates by a few lines the 3 page limit (see https://iclr.cc/Conferences/2018/CallForWorkshops). Please send us a fixed version of your PDF at iclr2018.programchairs@gmail.com by the end of Monday, February 19th, or else we will reject your paper.

Thanks,
ICLR2018 Program Chairs

---

> ### Public Comment · ~Anna_Thomas1 · 2018-02-18
> **Fixed length**
>
> Thank you for letting us know. We have sent in an updated version.

---

### Decision · Program_Chairs · 2018-03-20
**ICLR 2018 Workshop Acceptance Decision**

**Decision:**

Accept

**Comment:**

Congratulations, your paper was accepted to the ICLR workshop.